# Environmental Sustainability and Economic Growth in Greenland: Testing the Environmental Kuznets Curve

**Javier Arnaut** [1],* and **Johanna Lidman** [2]

1  Institute of Social Science, Economics, and Journalism, University of Greenland/Ilisimatusarfik, Manutooq 1, 3905 Nuussuaq, Greenland
2  Department of Economics, University of Aberdeen, Aberdeen AB24 3FX, UK; j.lidman.15@aberdeen.ac.uk
*  Correspondence: jaar@uni.gl; Tel.: +299-38-5629

**Abstract:** The environmental Kuznets curve (EKC) hypothesis assumes there is an inverted U-shape relationship between pollution and income per capita, implying an improvement in environmental quality when a growing economy reaches a high level of economic development. This study evaluated empirically the existence of the environmental Kuznets curve in Greenland for the period 1970–2018. Using an autoregressive distributed lag (ARDL) approach, the results show evidence of a U-shaped EKC in Greenland instead of the hypothesized inverted U-shape. The findings indicate that Greenland had initially experienced a decoupling transition during an early development stage associated with structural conditions of a small subsistence economy. However, once the country began to expand its industry, the trend began to reverse, creating a positive and significant relationship between $CO_2$ emissions and GDP per capita that is potentially detrimental to the Arctic natural environment.

**Keywords:** economic growth; environmental Kuznets curve; Greenland





## 1. Introduction

The continuous disintegration of the Greenland ice sheet and the rapid rise of global sea levels has generated a global concern on Greenland's changing Arctic environment. Despite the public awareness of the likely pervasive effects on the Arctic natural landscape, climate change is not a pressing concern within Greenland. The potential untapping of geological Arctic zones due to warmer temperatures is seen instead as an opportunity to generate local development out of extractive activities [1]. Promoting economic growth as a channel for attaining further financial self-sufficiency has been a key policy goal for the country, particularly after the historic achievement in 2009 of establishing its self-government. Nonetheless, although sustained growth rates have been elusive due to the country's strong reliance on volatile fishery commodities, a variety of measures are currently underway to encourage sustained growth through sectoral diversification, such as large-scale investments in mineral mining projects and transportation infrastructure for tourism.

Local community leaders, environmentalists, and scholars have begun to question whether the promotion of new industrial projects that intend to deliver growth could be detrimental to the country's fragile natural environment. Given that country's Arctic ecosystem has been largely affected by global-scale climate change [2,3], there are growing concerns about the risks of amplifying the existing environmental impact of climate change by compounding it with local greenhouse gases that emanate from the country's recent wave of urbanization and industrialization, e.g., [4,5]. To unravel the concerns on sustainability, it is central to examine whether Greenland has managed to "decouple" its economic growth process, that is, to raise the country's material wealth without considerable pressure and degradation of the local natural environment.

Measuring the environmental sustainability of Greenland through the pollution-growth nexus is pivotal because of the possibility of reaching a scenario where economic

growth becomes favorable for environmental quality, or what is known as the hypothesis of the environmental Kuznets curve (EKC). This premise is a theoretical and empirical observation that describes a process where a country in the early stages of development with low-income levels experiences an increase in environmental degradation as the country's income grows. The increase in degradation continues until the point when the country reaches a high level of development where income growth and environmental degradation exhibit an inverse relationship (i.e., income per capita increases and pollution decreases). The latter is usually associated with social and technological change after surpassing a threshold where countries invest in cleaner sources of energy generation, and there is an increase in public awareness (through a change in consumption behavior) promoting a better quality of the natural environment. As a result, it is hypothesized there is a long-term relationship between economic development and environmental degradation that takes the form of an inverse parabola or an inverted U-shape curve.

Since the seminal work by Grossman and Krueger [6], numerous empirical studies have addressed empirically the existence of the EKC [7]. Generally, results have been mixed and dependent on different modeling techniques and datasets for a variety of countries and regions. However, despite the importance and potential policy implications of these analyses for the Arctic region, no study has formally explored empirically the EKC hypothesis for the case of Greenland. This article aimed to fill this gap making a two-fold contribution to the literature: (i) The relationship between economic growth and $CO_2$ emissions was investigated by testing the EKC hypothesis for the period 1970–2018, and (ii) Energy usage and urbanization were included as determinants of $CO_2$ emissions in an augmented carbon emission function based on an ARDL (autoregressive distributed lag) approach.

The remainder of this paper is organized as follows. The next section briefly explains the Greenlandic context, followed by a section that contains a brief review of previous studies. Subsequently, the empirical model, the data, and the ARDL approach are presented. The last two sections analyze the empirical findings and provide some implications and concluding remarks.

## 2. The Context of Environmental Degradation in Greenland and Related Studies on the EKC in the Arctic

The continuous thaw of the Greenland ice sheet has been one of the benchmarks for measuring the global impact of climate change. It has been estimated that the current annual melt rate of the ice sheet has accelerated nine times since 1992 [8]. The IPCC (Intergovernmental Panel on Climate Change) concluded that by the end of the 21st century the average temperature increase in the Arctic will be about twice that of the increase of the global average, and during wintertime, Arctic ocean temperatures will rise more than three times faster [9,10]. If the current rate continues, the permanent effects are expected to be catastrophic not only for the Arctic but for most of the world's ecosystem. In the process, extreme natural events such as floods, wildfires, and droughts are likely to occur more frequently. Higher sea levels will have a direct impact on the living conditions of nearly half of the world's population that live in coastal zones [11]. Consequently, territorial displacement, involuntary migration, and social conflict are likely to emerge as outcomes of permanent environmental change [12].

There is unequivocal evidence that the rise of anthropogenic drivers of greenhouse gases has pushed upward the temperatures in the Arctic [13]. These changes have generated a significant loss of coastal sea ice and permafrost thawing in Greenland. The permafrost stores $CO_2$ from an ancient organic matter of decayed plants and animals, and hence a continuous thaw of it amplifies significantly the release of $CO_2$ emissions and methane, creating a feedback (known as a "positive feedback loop") that compounds the initial warming, adding yet more greenhouse gas to the atmosphere [14].

While warmer temperatures in the Arctic are likely to uncover areas rich in mineral deposits offering more opportunities for resource exploitation [15], there is no certainty yet of the "spillover" effect on the Greenlandic economy. To date, changes in coastal

sea ice thickness and permafrost thawing have represented a challenge to the traditional Greenlandic way of living. Greenland has a sparse population of around 56,000 inhabitants across small coastal towns and Nuuk, the country's capital that hosts nearly one-third of the total population. The majority of coastal communities in the country are heavily reliant on subsistence economic activities such as fishing and hunting. Because of the changes in ocean circulation, ice thickness, and ocean acidification in the Arctic [16], many of these communities that depend on winter routes for the timely re-supply of basic products have started to experience uncertainty by the changes in transportation routes [17]. Hence, shifting the economic structure out of the extant local traditional activities is becoming an unintended transformation of the local communities.

Authorities of the Arctic region have recognized the urgency for mitigating this phenomenon. The so-called Nuuk Declaration of 2011 was officially adopted by country members of the Arctic Council looking to facilitate regional policy cooperation to minimize the human and environmental impacts of climate change. Greenland has been effective in promoting renewable energy by building hydropower infrastructure since the early 1990s. Currently, there are five hydroelectric power plants in the country operated by the state-owned company Nukissiorfiit. The existing energy infrastructure in combination with a low-population density places Greenland among the world leaders of renewable energy, where 62.3% of its domestic electricity comes from hydropower [18]. Nonetheless, aside from Iceland, the current trends of $CO_2$ emissions in Greenland are still relatively higher than other Nordic countries (see Figure 1) with a similar large infrastructure of renewable energy sources.

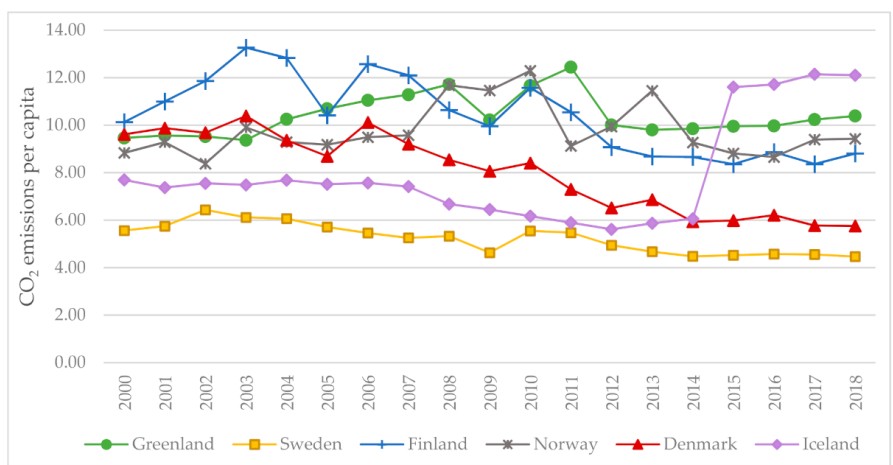

**Figure 1.** Carbon dioxide emissions ($CO_2$) per capita in Nordic countries since 2000. Source: See details in the data section in this article. For other Arctic countries, the data were retrieved from the World Bank Development Indicators Database until the year 2014 and linked to a comparable time series from the European Commission from 2015 to 2018.

However, the $CO_2$ interrelations with economic growth and the shape of the EKC are not directly observable across time because total carbon emissions might be driven by the interaction of a variety of factors, such as energy production and/or other relevant factors described in subsequent sections of this article. The relatively "high" carbon footprint in Greenland is related to the emissions of energy usage from fuel combustion from the economic activity in larger towns such as the Greenlandic capital, Nuuk, where most of the maritime shipping, aerial transportation, and the service industry takes place. Moreover, the dependence on the private supply of electricity and heating from power generators using imported diesel in several small towns and settlements outside the capital (and other medium-sized towns, such as Sisimiut and Ilulissat) contributes greatly to the national estimate of total $CO_2$ emissions in Greenland.

*Previous Studies of the EKC in the Arctic*

Empirical studies on the EKC in the Arctic region are relatively scarce. Some of the few empirical approximations are found in Baek [19] and other works such as Urban and Nordensvärd [20] and Olale et al. [21]. These analyses have used a sample of countries adjacent to or located within the Arctic region. The lack of disaggregated statistical information, issues with data consistency on regional output, and availability of data on different environmental pollutants are usually the reasons for the limited number of specific studies on the Arctic relative to other regions and countries.

On the other hand, there is a vast international body of literature covering a wide range of countries. The original idea of the EKC stems from the "bell-shaped" or "inverted U" relationship between income inequality and per capita income that Simon Kuznets [22] argued existed in the long run. Although it was the study of Panayotou [23] that suggested that an inverted-U relationship between pollution and income can be interpreted as an (environmental) Kuznets curve pattern, the work of Grossman and Krueger became seminal since it was one of the first data-driven analyses to test the existence of the growth-pollution nexus. Through a cross-section examination of 42 countries, the influential study analyzed the environmental impact of a free trade agreement between the United States and Mexico. Their analysis found that economic growth tended to alleviate pollution problems, and liberalization policies facilitated the process.

Several theoretical and empirical reevaluations emerged subsequently. Arrow et al. [24] argued that income is not exogenous to environmental degradation. It implied that rapid growth in an early stage of development could increase environmental degradation, generating an irreversible impact on subsequent growth, making higher levels of economic activity unsustainable. Stern [25] and several other empirical studies have also criticized the EKC on statistical grounds. Many of these empirical reassessments have pointed out the sensitivity of the existing results to the type of pollution variables used in the country sample, different econometric specifications, and modeling techniques. Although there is mixed evidence (country-specific) confirming the existence of the EKC globally, the relationship across countries is statistically weak.

Generally, countries within the Arctic region have been excluded from samples in large cross-country analyses. The Arctic region comprises parts of Northern Canada, Alaska, Finland, Greenland, Iceland, Norway, Russia, and Sweden. In a comprehensive analysis, the abovementioned study by Baek examined the EKC in an individual-country sample that included Canada, Finland, Denmark, Iceland, Norway, Sweden, and the United States from 1960 to 2010. The analysis, however, did not include Greenland. Through a cointegration empirical framework, Baek's study found weak evidence of the EKC hypothesis in the Arctic region. Economic growth had only a significant and positive impact on $CO_2$ emissions in a few of the countries (for example, Denmark and Iceland). Still, the study found that energy consumption has had a significant adverse impact on the environment in the majority of the Arctic countries (excluding Greenland).

The study by Urban and Nordensvärd using a sample of Scandinavian countries between 1960 to 2015 analyzed various data trends related to the EKC and sketched graphically some functional relationships. Using data on $CO_2$ emissions per capita and GDP per capita, their descriptive study outlined the existence of the EKC in Sweden, Denmark, Finland, and Iceland, yet their results did not employ statistical testing to verify the significance and consistency of the evidence. On the other hand, Olale et al. explored the ECK for the Canadian provincial/territorial areas that included the Arctic region for the period 1990–2014. Their work found confirmatory statistical evidence of the EKC hypothesis using a fixed-effects regression framework. Their estimates found that the EKC existed at the national level but interestingly only in half of the provinces and territories. It was found in particular that the EKC pattern was statistically absent in the Arctic territory of Nunavut. The authors pointed out that technology differences and other specific local characteristics of energy usage may play a role in determining the pollution-growth nexus in Arctic Canada.

The general message that available studies evaluating the EKC for the Arctic region is that economic growth has had a positive impact on the environment in a very specific case (i.e., Iceland), while in more natural-resource dependent countries/regions (e.g., Norway and Northern Canada) it has not. This is related to the different experiences of low carbon energy transitions, where technological change has allowed an increasing share of hydropower, wind, and solar energy sources. This has played a determining role in shaping the trends of $CO_2$ emissions per capita and forming ultimately the EKC. However, a problem arises in natural resource-based countries/regions that are bounded and over-reliant on polluting tradable (e.g., oil and mining) and non-tradable sectors (e.g., construction). Thus, the challenges that these natural-resource regions face are not only related to technological change but to the development of institutional capabilities and policies for the acceleration of structural change toward a low-carbon tertiary sector [26,27].

## 3. Data, Model, and Empirical Method

### 3.1. Data Sources

The most frequently used variables to measure pollution in EKC studies are the emissions of carbon dioxide, sulfur dioxide, methane, and other types of industrial and non-industrial concentrations of greenhouse gases. For the case of Greenland, this study used carbon dioxide emissions since they are considered by major international institutions as the primary greenhouse gas in the atmosphere, accounting for more than three-quarters of the total global emissions [28] and 94% of Greenland's total greenhouse concentrations [29] (p. 637).

The period of the analysis covered the years from 1970 to 2018. The span of the sample was determined by the availability of the data sources. The information on total carbon dioxide emissions refers to metric kilotons (kt) of carbon dioxide stemming from the burning of fossil fuels and the consumption of solid, liquid, and gas fuels and gas flaring. The data were obtained from the World Bank Development Indicators Database [30] for the years 1970 to 2014. The last four observations (for the years 2015 to 2018) were from Statistics Greenland Databank [31], originally compiled from Denmark's National Inventory Report [29], which are a comparable time series that follow the estimation methods under the IPCC guidelines. The information on total carbon dioxide emissions was adjusted into per capita terms using annual data of the total country's population from Statistics Greenland.

Data for gross domestic product (GDP) per capita in constant 2010 US dollars and the variable measuring urbanization (urban population as a percentage of the total population) for 1970 to 2018 were both retrieved from the World Development Indicators database. Given that official data on energy consumption for Greenland are only available (and standardized) for recent years, the analysis followed other studies employing long-span data of electricity production as a proxy for energy usage in modeling the EKC, e.g., [32]. The information on energy usage refers to the total electricity production measured in millions of kilowatt-hours (kWh), and it was retrieved from Greenland Statistical Yearbooks (Grønland Statistisk Arbog) for the corresponding years. The descriptive statistics of all the variables employed in the analysis are presented in Appendix A (Table A1).

### 3.2. Model

The model in this study followed a related functional specification as that taken by Baek [19] for a sample of Arctic countries and by Azam and Khan [33] for a larger sample of countries with different levels of development. The long-run relationship is specified as follows:

$$CO_{2_t} = \int \left( Y_t,\ Y_t^2,\ En_t, Urb_t \right) \tag{1}$$

where $CO_2$ are total carbon emissions per capita; $Y_t$ represents per capita real income, which is measured as real GDP per capita; $Y_t^2$ is the square of per capita real income; $En_t$ is total electricity production; and $Urb_t$ is the urban population as a share of the total

population. It is important to indicate that aside from the functional relationships, the variables of electricity production and urbanization are also included in Equation (1) to minimize econometric issues such as spurious regression and the omitted variable bias that commonly arise in time series models [25]. Rewriting Equation (1) in natural logarithms and including a random error term $\varepsilon_t$ that includes other possible causes of environmental degradation and a constant $\beta_0$, the equation to estimate is as follows:

$$logco_{2_t} = \beta_0 + \beta_1 log(y_t) + \beta_2 log(y_t)^2 + \beta_3 log(en)_t + \beta_4 log(urb)_t + \varepsilon_t \tag{2}$$

The primary coefficients to test are $\beta_1$ and $\beta_2$. The EKC hypothesis suggests a $\beta_1 > 0$ and $\beta_2 < 0$, implying the inverted U-shape or a parabolic pattern. Again, the significance of these coefficients with these expected signs would be supporting evidence for the EKC hypothesis. However, it is possible that depending on the sign of these $\beta$ coefficients (related to income $y_t$ and $y_t^2$), the relationship between $CO_2$ and income adopts different shapes, such as the following:

$\beta_1 = \beta_2 = 0$, flat shape or no relationship between $CO_2$ emissions and income.
$\beta_1 > 0$ and $\beta_2 < 0$, there is an inverted U-shape between $CO_2$ emissions and income (the EKC).
$\beta_1 < 0$ and $\beta_2 = 0$, a monotonic decreasing relationship between $CO_2$ emissions and income.
$\beta_1 < 0$ and $\beta_2 > 0$, there is a U-shaped relationship between $CO_2$ emissions and income.

Regarding additional explanatory variables that might affect the environmental quality, if $\beta_3 > 0$ it would imply that an increase in energy usage results in a rise in $CO_2$ emissions. As for $\beta_4$, although in most cases due to the scale effect of an increasing population density, urbanization tends to increase pollution ($\beta_4 > 0$). However, the effect can also be inverse because urbanization could mitigate $CO_2$ emissions if in urban areas there is access to cleaner energy sources, e.g., [34]. This feature could be the case for Greenland, where several small coastal towns rely on electricity from fossil fuels, while in relatively larger towns or urban areas, hydropower electricity—a "cleaner" energy source—is available. Therefore, it is possible to obtain a $\beta_4 < 0$.

### 3.3. Empirical Method

To examine the aforementioned relationships, and given that the EKC is regarded as a long-term phenomenon, the analysis drew on the auto-regressive distributed lag (ARDL) approach suggested by Pesaran et al. [35]. A key advantage of this empirical view (also known as bounds testing) is the circumvention of error serial correlation because the dependent and explanatory variables are also endogenous in the model. It also allows for both nonstationary time series as well as for times series with a different order of integration (e.g., variables may be stationary in levels or the first difference). A representation of the ARDL approach using the variables in the model previously presented can be expressed as follows:

$$\Delta co_{2_t} = \alpha_0 + \sum_{i=1}^{p} \alpha_{1i}\Delta co_{2_{t-1}} + \sum_{i=0}^{p} \alpha_{2i}\Delta y_{t-i} + \sum_{i=0}^{p} \alpha_{3i}\Delta y_{t-i}^2 + \sum_{i=0}^{p} \alpha_{4i}\Delta en_{t-i} + \sum_{i=0}^{p} \alpha_{5i}\Delta urb_{t-i}$$
$$+ \alpha_6 co_{2_{t-1}} + \alpha_7 y_{t-1} + \alpha_8 y_{t-1}^2 + \alpha_9 en_{t-1} + \alpha_{10} urb_{t-1} + \mu_t \tag{3}$$

As long as it can be assumed that the error term $\mu_t$ follows a white noise process, then Equation (3) can be estimated consistently as a linear combination of the lagged level and differentiated variables. The parameters $\alpha_{1i}$, $\alpha_{2i}$, $\alpha_{3i}$, $\alpha_{4i}$, and $\alpha_{5i}$ in Equation (3) represent the short-run parameter dynamics, and $\alpha_6$, $\alpha_7$, $\alpha_8$, $\alpha_9$, and $\alpha_{10}$ are the long-run parameters. According to the bounds test, the null hypothesis, which implies no cointegration (absence of a long-run relationship), is $H_0$: $\alpha_6 = \alpha_7 = \alpha_8 = \alpha_9 = \alpha_{10} = 0$, contrarywise to the alternative hypothesis, which is $H_1$: $\alpha_6 \neq \alpha_7 \neq \alpha_8 \neq \alpha_9 \neq \alpha_{10} \neq 0$. The model was tested based on the joint *F*-test (Wald statistic). The critical values for the bound test are reported in the appendix of Pesaran et al. [35].

In case of establishing cointegrating relationships among the variables, it is then possible to estimate a long-run model using the following equation representation:

$$\Delta co_{2_t} = \delta_0 + \sum_{i=1}^{p} \delta_{1i} \Delta co_{2_{t-1}} + \sum_{i=0}^{p} \delta_{2i} \Delta y_{t-i} + \sum_{i=0}^{p} \delta_{3i} \Delta y_{t-i}^2 + \sum_{i=0}^{p} \delta_{4i} \Delta en_{t-i} + \sum_{i=0}^{p} \delta_{5i} \Delta urb_{t-i} + \lambda_1 EC_{t-1} + \nu_t \quad (4)$$

In Equation (4), $EC_{t-1}$ refers to the error correction term, and $\lambda_1$ is the parameter that measures the speed of adjustment of the variables to the long-run equilibrium. The linear specification of the ARDL model assumes the existence of symmetrical effects (i.e., excludes nonlinearities) of the explanatory variables on the dependent variable. Some recent panel data studies employing nonparametric estimators, e.g., [36], have found that this assumption (linear specification) could generate issues of model misspecification and omitted variable bias. However, assuming normality in the residuals of the abovementioned functional relationship and aiming to retain a parsimonious estimation, we considered the parametric linear estimation approach.

## 4. Results and Discussion of Findings

### 4.1. Empirical Results

Results from unit root tests are presented in Table 1 and are relevant to determining the presence of stochastic stationarity in the data. According to the tests, except for $CO_2$, the series were nonstationary at level (i.e., contain unit roots) and became stationary at first difference. Thus, most of the variables were integrated of order one. As mentioned previously, the ARDL approach allows variables to be either integrated into one order or mixed order; hence, it was suitable to model the relationship dynamics of the $CO_2$ variable with the stochastic presence of stationarity of the variable at level.

**Table 1.** Unit Roots Tests: Augmented Dickey–Fuller (ADF) and Philips–Perron (PP).

| Test | Levels | | First Difference | |
|------|-----------|-------------|-----------|-------------|
| | Variable | t-Statistic | Variable | t-Statistic |
| ADF | log ($CO_2$) | −6.6819 *** | log ($CO_2$) | −7.6427 *** |
| | log (y) | −2.3356 | log (y) | −6.1005 *** |
| | log (y)$^2$ | −2.2021 | log (y)$^2$ | −6.1233 *** |
| | log (en) | −2.2266 | log (en) | −7.7585 *** |
| | log (urb) | −0.8605 | log (urb) | −5.4788 *** |
| PP | log ($CO_2$) | 6.6824 *** | log ($CO_2$) | −12.8704 *** |
| | log (y) | −2.5767 | log (y) | −6.1593 *** |
| | log (y)$^2$ | −2.4680 | log (y)$^2$ | −6.1842 *** |
| | log (en) | −2.1086 | log (en) | −12.9354 *** |
| | log (urb) | −2.0215 | log (urb) | −5.4989 *** |

Note: The null hypothesis was that the variable had a unit root. Significance at the 1% level is denoted by ***. In both tests, the models were considered as the models with intercept and trend. The Schwarz information criterion was used to select the lag length. $CO_2$: total carbon emissions per capita; y: per capita real income; y$^2$: square of per capita real income; en: total electricity production; and urb: urban population as a share of the total population.

Various information criteria methods were used to select the optimal lag length for the model estimation of the long-run model (Appendix A, Table A2). The criteria indicated that one lag (one year) was the optimal lag length structure to examine the long-term relationship among the variables.

Following the methodology, to establish a long-term relationship (cointegration) among the series, a bounds test was applied to test the model (Equation (3)), and the results are presented in Table 2. Given that the F-statistic (21.83) of the estimated model was higher than the upper critical values at 1%, it implies the rejection of the null hypothe-

sis of no long-run relationship, and thus confirming the existence of cointegration among the variables.

**Table 2.** Auto-Regressive Distributed Lag (ARDL) Bounds Test (Cointegration Relationship).

| Estimated Model | F-Statistic | 1% Critical Values | |
| --- | --- | --- | --- |
| | | I(0) | I(1) |
| $CO_2$ = f (y, $y^2$, en, urb) | 21.83 | 3.74 | 5.06 |

Note: The null hypothesis was that there was no levels relationship. The critical values were reported by Pesaran et al. The selected model was ARDL (1, 1, 0, 0, 0) with a constant and no trend.

The estimates of the long run and short model are presented in Table 3. The results of the long term estimates display a negative and significant coefficient on the income variable ($-18.33$) and significant and positive for the squared income term. (Alternates to the quadratic function were also considered. For example, a cubic term of the income per capita variable was included. However, no statistical significance was found, and the functional specification test confirmed the misspecification if the cubic term was included in the model). A similar result was shown for the short-run estimates. These results reveal that the relationship between $CO_2$ and income followed a U-shape (as stated earlier in different model specifications $\beta_1 < 0$ and $\beta_2 > 0$). The coefficients on income were relatively large, implying that an increase in 1% of income represented an 18.33% reduction of $CO_2$ emissions (roughly 20% in the short run) (The turning point of the curve (long-run coefficients) was $\exp(-\beta_1/2\beta_2) \cong US\$41,913$). On the other hand, the coefficients (short and long run) on energy usage (en) were significant and positive (0.29), indicating the scale effect of economic growth that materialized in the monotonic positive effect of energy consumption on $CO_2$ emissions.

**Table 3.** ARDL Model: Long-Run and Short-Run Estimates.

| Dependent Variable: log($CO_2$) | | |
| --- | --- | --- |
| ARDL structure (1, 1, 0, 0, 0) | | |
| Long-run | | |
| **Regressors** | **Coefficient** | ***t*-value** |
| log(y) | $-18.3368$ | $-3.7458$ *** |
| log(y)$^2$ | 0.9130 | 3.8367 *** |
| log(en) | 0.2928 | 2.1446 ** |
| log(urb) | $-5.7150$ | $-2.7070$ *** |
| **Diagnostics** | **tests-stats** | ***p*-value** |
| **Serial autocorrelation** | 0.8271 | 0.4462 |
| **Heteroskedasticity** | 1.4328 | 0.2069 |
| **Functional specification** | 0.3035 | 0.5851 |
| Short-run | | |
| **Regressors** | **Coefficient** | ***t*-value** |
| $\Delta$log(y) | $-20.4627$ | $-3.6640$ *** |
| $\Delta$log(y)2 | 1.0189 | 3.7419 *** |
| $\Delta$log(en) | 0.3268 | 2.1243 ** |
| $\Delta$log(urb) | $-6.3766$ | $-2.6285$ *** |
| Error correction term (EC) | $-1.1159$ | $-11.0133$ *** |

*Note*: Year binary variables (as fixed regressors) and an intercept were included in the estimation. Significance at 1% and 5% level is denoted by *** and **. The serial correlation test refers to the Breusch–Godfrey Serial Correlation LM Test. To test for heteroskedasticity, the Breusch–Pagan–Godfrey test was used. The Ramsey RESET test was used to examine whether the model was correctly specified.

The results also show a negative and significant coefficient of the urbanization variable (urb) in the short and long run. Although it may appear counterintuitive, greater urbanization in Greenland was associated with an improvement of environmental quality (a reduction of $CO_2$ emissions). The population movement out of settlements into larger towns (particularly to Nuuk) that rely less on fossil fuel electricity (e.g., imported diesel) for households and more on hydropower electricity is a characteristic of Greenlandic urbanization over the last three decades. In this case, a 1% increase in the rate of urbanization generated roughly 5.7% less $CO_2$ emissions (long-term model).

The negative coefficient of error correction term (EC) in the short-run equation indicates the adjustment speed of the variables toward the long-run equilibrium. The model's diagnostics showed that the estimates were consistent and stable over the period studied. Figure A2 in the Appendix A displays that the cumulative sum of recursive (squared) residuals were within the critical bands (5% confidence intervals), confirming the stability of the parameters of the model.

### 4.2. Discussion of Results and Implications

The original EKC hypothesis postulated in seminal studies [6,23] suggests that at the early stages of development, industrialization and urbanization deplete the natural environment, raising $CO_2$ emissions. In this stage, economic growth and $CO_2$ emissions are positively related to each other (i.e., $\beta_1 > 0$). The process continues until a point where technological change emerges, promoting the use of low-carbon energy sources and an expansion of the service sector resulting in a reduction of $CO_2$ emissions ($\beta_1 > 0$ and $\beta_2 < 0$). The estimates presented in this study do not support the EKC hypothesis of the existence of an inverted-U shape relationship between economic growth and environmental quality in Greenland. Instead, the model's estimates show the opposite ($\beta_1 < 0$ and $\beta_2 > 0$): there was a U-shaped relationship implying that the level of $CO_2$ emissions initially decreased as income rose, reaching a bottom or stabilizing point, and then $CO_2$ emissions increased as income grew. The results suggest that during an early development stage, Greenland had "decoupled" economic growth (increasing per capita income while reducing $CO_2$ emissions) until a stage where, as a consequence of industrialization, energy usage began to push greenhouse gasses emissions upward, moving in the same direction as the growth of per capita income.

The question of whether there was a decoupling process in Greenland in an early development stage is related to the historical structure of the economy. Most of the Greenlandic communities were founded on small formal and informal sectors of traditional subsistence and commercial and noncommercial activities from hunting and fishing. The majority of these activities are typically carbon neutral. However, industrialization and the ensuing increase of aerial and maritime transportation into the country began to reverse the decoupling effect.

The impact of the increase in electricity production had an expected adverse effect on environmental quality in Greenland. Although there was a clear energy transition toward hydropower electricity, several small towns still rely on gas–oil, gasoline, and kerosene, which are significant contributors to the emissions of $CO_2$. Urbanization, however, as the estimates presented indicate, has been beneficial to the reduction of $CO_2$ emissions according to the estimates. This effect is associated with structural change reflected in the population movement (intra-regional migration) into larger towns where hydropower electricity is available.

Overall, the results of a U-shape curve in Greenland are in line with other studies that have analyzed regions with economies that are natural resource-intensive, e.g., [26,27]. The literature on natural resource dependence and the so-called "Dutch disease" stresses how countries with a structural overreliance on economic activities involving (polluting) natural resource-based sectors generally experience lower economic growth in comparison to countries with less dependence on those activities [37]. This represents a climate change mitigation challenge for policymakers in Greenland, given that in the search of sectoral

diversification, investments in highly energy-intensive sectors such as mineral mining (for example, rare-earths, lead, and nickel) and construction (roads and airports) are the apparent path to attain sustained economic growth rates and financial self-sufficiency.

A reconversion toward the primary sector, that is, a return to a previous development stage characterized by a small subsistence economy based on hunting and fishing, will likely not deliver sustained economic growth for Greenland. Consequently, it would delay the policy goal of achieving greater financial self-sufficiency, creating a local policy dilemma. Although economic growth is a legitimate imperative for financial self-sufficiency, it can also be at the expense of Arctic environmental degradation by compounding the existing and pervasive effects of climate change.

Nonetheless, the global economy is transitioning to the production of low-carbon emission energy devices and infrastructure [38]. Greenland can capitalize on this global energy transition because it possesses some of the natural resources (for example, rare-earths) to build the new low-carbon emission technology. Additionally, the nascent local mining industry in Greenland could be encouraged to utilize the new and low-cost extraction methods that contribute to the local and international decarbonization of the metal supply chains, e.g., [39]. The inability (or postponement) in promoting sustainable economic development using cleaner technologies in energy-intensive sectors (mineral extraction and construction) to mitigate the environmental impact of growth will likely entail a continuation of the relationship (the U-shape) between per capita income and $CO_2$ emissions, as found in the present study.

## 5. Conclusions

The primary objective of this study was to test the environmental Kuznets curve for Greenland throughout 1970–2018. The EKC hypothesis was examined under the ARDL framework by using per capita carbon dioxide ($CO_2$) emissions as an indicator of environmental quality, total electricity production as a measure of energy usage, and a variable accounting for the country's urbanization. The empirical results show a statistically significant long- and short-run relationship between the variables. However, the results do not support the existence of the inverted U-shape hypothesized by the EKC. The significant and negative sign of income, together with a positive sign of the quadratic term of income, shows instead that there is a U-shape pattern in the relationship between $CO_2$ and economic growth. The model's diagnostics and stability confirm the reliability of the results.

This study contributes to the literature on EKC for the case of Greenland by establishing the type of dynamic relationship between economic growth and $CO_2$ emissions across time, unraveling the recent concerns on whether the country has managed to decouple its growth process. The results indicate that Greenland had initially experienced a decoupling transition ($CO_2$ emissions per capita were not statistically related to GDP per capita) during an early development stage associated with structural conditions of a small subsistence economy. However, once the country began to expand its industry (commercial fishing, retail, and transportation), the trend began to reverse, creating a positive and significant relationship between $CO_2$ emissions and GDP per capita that is potentially detrimental to the Arctic natural environment. The results are in line with other recent quantitative studies on natural resource-dependent economies, where their over-reliance on natural resource sectors ultimately have determined the shape of the curve [26,27].

Given the vulnerability of the Arctic ecosystem, as shown by several environmental studies and the assessment reports by the IPCC, e.g., [40], the local authorities have to continue strengthening and framing special policies for renewable energy by levying taxes on fossil fuels and subsidizing renewable energies. The results of this study do not necessarily imply a halt to the current sectoral diversification efforts (for example, toward mining or/and tourism). The promotion of sustainable economic development that mitigates the increase of greenhouse emissions with the use of cleaner technologies and modes of production on the newly promoted activities will play a key role in reversing the current shape of the curve.

There is a caveat in the analysis related to the availability of data sources that is worth noting. Greenland's energy transition toward renewable sources has been an important development that counteracts the detrimental effects of industrialization on the Arctic natural environment. However, the statistical records on hydroelectric power generation have only been compiled for recent years. Future studies on Greenland should incorporate a measure of the share of renewable energy usage to accurately quantify the positive effect (if any) of moving away from fossil fuel combustion at the household and industry level. A distinction of the type of energy usage would provide a clearer and more accurate indication of whether there is a reversal of the current trend after following an environmentally sustainable pattern.

**Author Contributions:** Conceptualization, J.A.; methodology, J.A.; data curation, J.A. and J.L.; writing—original draft preparation, J.A. and J.L.; writing—review and editing, J.A. and J.L. All authors have read and agreed to the published version of the manuscript.

**Funding:** This research received no external funding.

**Institutional Review Board Statement:** Not applicable.

**Informed Consent Statement:** Not applicable.

**Data Availability Statement:** Data sharing not applicable. No new data were created or analyzed in this study. Data sharing is not applicable to this article.

**Acknowledgments:** The authors acknowledge useful comments and suggestions from participants of the research seminar at the University of Greenland in September 2020. The usual disclaimer applies.

**Conflicts of Interest:** The authors declare no conflict of interest.

## Appendix A

**Table A1.** Descriptive Statistics.

|  | $CO_2$ | y | en | urb |
|---|---|---|---|---|
| Mean | 9.76 | 30,565.06 | 201.80 | 79.87 |
| Median | 9.94 | 27,499.38 | 164.50 | 80.50 |
| Maximum | 14.62 | 47,974.07 | 343.15 | 86.33 |
| Minimum | 4.34 | 15,656.47 | 54.20 | 72.74 |
| Std. Dev. | 1.80 | 8715.16 | 96.94 | 3.89 |
| Skewness | −0.83 | 0.48 | −0.03 | −0.16 |
| Kurtosis | 5.59 | 2.22 | 1.34 | 1.94 |
| Observations | 47 | 47 | 47 | 47 |

**Table A2.** Results of Optimal Lag Length Criteria.

| Lag | LR | FPE | AIC | SC | HQ |
|---|---|---|---|---|---|
| 0 | NA | 5.19E-11 | −9.492958 | −9.288168 | −9.417438 |
| 1 | 537.5831 [a] | 8.20e-17 [a] | −22.85944 [a] | −21.63070 [a] | −22.4063 [a] |
| 2 | 35.46262 | 9.10E-17 | −22.80486 | −20.55216 | −21.97413 |
| 3 | 26.34841 | 1.25E-16 | −22.61793 | −19.34128 | −21.40961 |
| 4 | 29.34337 | 1.38E-16 | −22.78893 | −18.48833 | −21.2030 |

Note: The lag order selected by the criterion is denoted by "[a]". LR: sequentially modified LR test statistic (each test at 5% level); FPE: final prediction error; AIC: Akaike information criterion; SC: Schwarz information criterion; HQ: Hannan–Quinn information criterion.

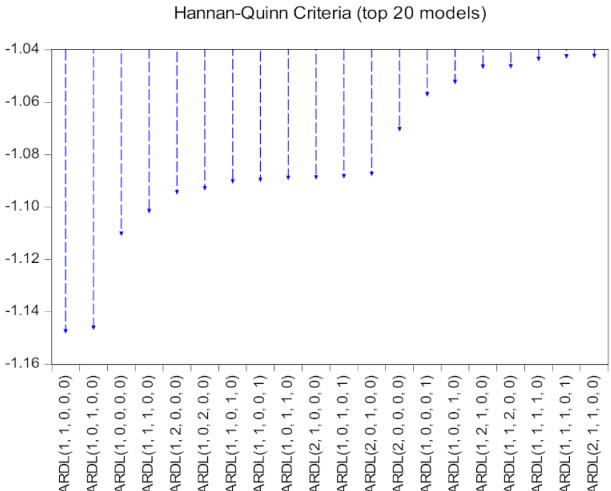

**Figure A1.** Summary of Model Selection for the ARDL Including All Variables.

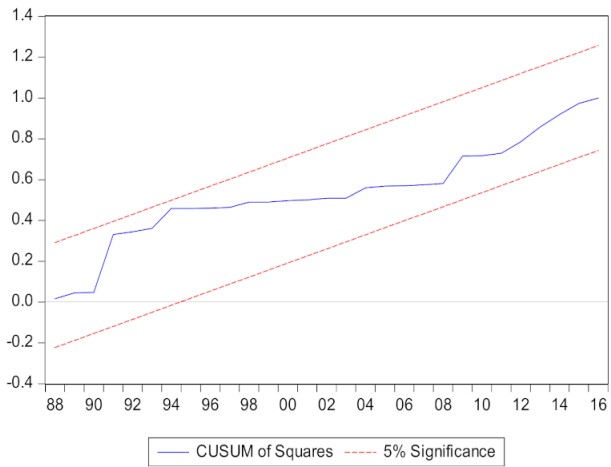

**Figure A2.** Coefficient Stability Test: Cumulative sum of Squares (CUSUMSQ).

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
