# Peer review of "Environmental Sustainability and Economic Growth in Greenland: Testing the Environmental Kuznets Curve"

_sustainability, doi:10.3390/su13031228_

Round 1

Reviewer 1 Report

The paper is interesting, it does not provide particularly innovative results but it is still worthy of publication. However, the authors should make some changes.

The introduction is too short. It does not make the research question attractive and it does not identify the line of literature in which the study conducted is inserted. Furthermore, I am not convinced of the choice to break the introduction into two subsections. This choice is contrary to the standards generally accepted by international journals.

As for the literature review, if the studies related to the Environmental Kuznets Curve in the Artic are really scarce, it is necessary to discuss them more. At present it is a simple examination of previous studies without any systematization.

Furthermore, it is necessary to identify a reference theory capable of framing the relationship under study.

The methodology and presentation of the results seem adequate, but I have many doubts about the discussion. It is in fact necessary to frame the results in the literature and explain them in line with the chosen theory. In the absence of this, the discussion is nothing more than a trivial presentation of the results obtained.

Finally, the conclusions must be totally rewritten. It is necessary, after having presented the objectives and the results obtained, to add the contributions that the study offers to the academic literature. Furthermore, the implications must be included. Finally, the authors should include the limitations of the study and suggestions for future research.

Good luck.

Author Response

The author(s):

We would like to thank the reviewer for spending time reading our paper and giving valuable comments and suggestions. We are grateful for making us reflect on some aspects of the paper and overall, for helping us to improve it. We have made careful considerations and revisions based on the comments. The following paragraphs summarize the changes made to the original manuscript.

Comment by the reviewer: I am not convinced of the choice to break the introduction into two subsections. This choice is contrary to the standards generally accepted by international journals.

Response by the author(s): We appreciate this observation. We originally followed some predetermined layout of other papers published by this journal. We were also not fully convinced about that format so thanks to this comment we have now changed it back as proposed by the reviewer. We have merged the introductory sections into a single one as a standard style of an "Introduction".

Comment by the reviewer: As for the literature review, if the studies related to the Environmental Kuznets Curve in the Artic are really scarce, it is necessary to discuss them more. At present, it is a simple examination of previous studies without any systematization.

Response by the author(s): We understand the comment and we appreciate for making us reflect on this. We agree on this. Yet, as mentioned, given the scarce number of studies on the EKC for the Arctic and the short content of these (in terms of theory and empirics on the case of the Arctic), it is was difficult to go beyond the existing literature and elaborate further (for instance, making a table of all the previous findings in the literature). However, to summarize and clarify these we have now written (at the end of subsection 2.1 "Previous studies of the EKC in the Arctic") an additional paragraph as follows:

“The general message that available studies evaluating the EKC for the Arctic region is that economic growth has had a positive impact on the environment in very specific cases (i.e., Iceland), while more natural-resource dependent countries/regions (e.g., Norway, and Northern Canada) has not. This is related to the different experiences of low carbon energy transitions where technological change has allowed an increasing share of hydropower, wind, and solar energy sources. This plays a determining role in shaping the trends of CO2 emissions per capita and forming ultimately the EKC. However, a problem arises in natural resource-based countries/regions that are bounded and over-reliant on polluting tradable (e.g., oil and mining) and non-tradable sectors (e.g., construction). Thus, the challenges that these natural-resource regions face are not only related to technological change but to the development of institutional capabilities and policies for the acceleration of structural change towards a low-carbon tertiary sector [26, 27].”

Comment by the reviewer: It is necessary to frame the results in the literature and explain them in line with the chosen theory.

Response by the author(s): To frame the results as suggested, we have re-written the initial paragraph in section 4.2. “Discussion of results and implications”.

“The original EKC hypothesis postulated in seminal studies [6,23] suggests that at the early stages of development, industrialization and urbanization deplete the natural environment raising CO2 emissions. In this stage, economic growth and CO2 emissions are positively related to each other (i.e. formula characters in the text). The process continues until a point where technological change emerges promoting the use of low-carbon energy sources and an expansion of the service sector resulting in a reduction of CO2 emissions (see formula characters in the text). The estimates presented in this study do not support the EKC hypothesis of the existence of an inverted-U shape relationship between economic growth and environmental quality in Greenland. Instead, the model’s estimates show the opposite (see formula characters in the text): there is a U-shape relationship implying that the level of CO2 emissions initially decreases as income rises, reaching a bottom or stabilizing point, and then CO2 emissions increase as income grows.”

Comment by the reviewer: The conclusions must be rewritten. It is necessary, after having presented the objectives and the results obtained, to add the contributions that the study offers to the academic literature. Furthermore, the implications must be included. Finally, the authors should include the limitations of the study and suggestions for future research.

Response by the author(s): We appreciate this suggestion. As recommended, we have re-written the conclusions. We have now added three new paragraphs in the conclusions which include the contribution to the literature, the implications, and the limitations (the paragraph explaining the limitations was relocated to the conclusions as suggested):

“This study contributes to the literature on EKC for the case of Greenland by establishing the type of dynamic relationship between economic growth and CO2 emissions across time, unraveling the recent concerns on whether the country has managed to “decouple” its growth process. The results indicate that Greenland had initially experienced a decoupling transition (CO2 emissions per capita were not statistically related to GDP per capita) during an early development stage associated with structural conditions of a small subsistence economy. However, once the country began to expand its industry (commercial fishing, retail, and transportation), the trend began to reverse creating a positive and significant relationship between CO2 emissions and GDP per capita that are potentially detrimental to the arctic natural environment. The results are in line with other recent quantitative studies on natural resource-dependent economies where their over-reliance on natural resource sectors ultimately have determined the shape of the curve [26, 27].”

“Given the vulnerability of the arctic ecosystem as shown by several environmental studies and the assessment reports by the IPCC, e.g., [40], the local authorities have to continue strengthening and framing special policies for renewable energy by levying taxes on fossil fuels and subsidizing renewable energies. The results of this study do not necessarily imply a halt to the current sectoral diversification efforts (for example towards mining or/and tourism). The promotion of sustainable economic development that mitigates the increase of greenhouse-emissions with the use of cleaner technologies and modes of production on the newly promoted activities will play a key role in reversing the current shape of the curve.”

“There is a caveat in the analysis related to the availability of data sources that are worth noting. Greenland’s energy transition towards renewable sources has been an important development that counteracts the detrimental effects of industrialization on the arctic natural environment. However, the statistical records on hydroelectric power generation have only been compiled for recent years. Future studies on Greenland should incorporate a measure of the share of renewable energy usage to accurately quantify the positive effect (if any) of moving away from fossil fuel combustion at the household and industry level. A distinction of the type of energy usage would provide a clearer and more accurate indication of whether there is a reversal of the current trend following an environmentally sustainable pattern.”

The author(s) (2021).

Reviewer 2 Report

The presented study is a very interesting, topical, and methodologically sophisticated contribution to the literature on the Environmental Kuznets Curve conjecture. Focusing on Greenland, a case study with a very special geographical and demographic context makes the study also relevant for environmental policies in the context of the Nordic countries in general and Greenland in particular. The study has a straightforward structure.

Please make the last sentence of the abstract explicit:"The article considers some economic implications of the empirical findings." Which implications are important?

The authors employ the usual ARDL methodology for their empirical assessment. This is a suitable methodology for the assessment of the EKC-hypothesis. Nevertheless, ARDL does not account for nonlinearities. There are methodologies such as nonlinear ARDL and nonparametric estimators that take account for this confinement of the linear ARDL models. To account for this limitation mention shortly the confinements of the linear and / or parametric estimators. These are especially model misspecification and omitted variable bias. To this end refer explicitly to Sadik-Zada & Loewenstein (2020). 

In the discussion part, row 326, the authors discuss the issue of the Dutch disease and resource curse in the respective context. They vindicate the finding of the U-shaped income-environment relationship referring to Badeeb et al. (2020). Nevertheless, there are also many other studies, which validate a minotonically increasing relationship. In addition, there is also a theoretical contribution, suggested in Socio-Economic Planning Sciences in a paper "Puzzle of Greenhouse Gas Footprints of Oil Abundance", which proposes a net theoretical explanation for a divergence from the inverted U-shaped relationship. Refer shortly but explicitly to this contribution in your paper. The authors attribute this to de-industrialization tendencies, whereby these tendencies contemplate not only Dutch disease but also other symptoms of the resource curse. To this end make use of the following diagram from Sadik-Zada, Loewenstein&Hasanli (2019) and A Note on Revenue Distribution Patterns and Rent-Seeking Incentive (2018):

Author Response

The author(s):

We would like to thank the reviewer for spending time reading our paper and giving valuable comments and suggestions. We are grateful for making us reflect on some aspects of the paper and overall, for helping us to improve it. We have made careful considerations and revisions based on the comments. The following paragraphs summarize the changes made to the original manuscript.

Comment by the reviewer.

Please make the last sentence of the abstract explicit: "The article considers some economic implications of the empirical findings." Which implications are important?

Response by the author(s): We thank the reviewer for making us see this issue on the abstract. We have now added a few lines that clarify what was found in the paper. We believe that the abstract is now more comprehensive by adding at the end of it the following:

“The findings indicate that Greenland had initially experienced a decoupling transition during an early development stage associated with structural conditions of a small subsistence economy. However, once the country began to expand its industry, the trend began to reverse creating a positive and significant relationship between CO2 emissions and GDP per capita that are potentially detrimental to the arctic natural environment.”

Comment by the reviewer:

In the discussion part, row 326, the authors discuss the issue of the Dutch disease and resource curse in the respective context. They vindicate the finding of the U-shaped income-environment relationship referring to Badeeb et al. (2020). Nevertheless, there are also many other studies, which validate a monotonically increasing relationship. In addition, there is also a theoretical contribution, suggested in Socio-Economic Planning Sciences in a paper "Puzzle of Greenhouse Gas Footprints of Oil Abundance", which proposes a net theoretical explanation for a divergence from the inverted U-shaped relationship. Refer shortly but explicitly to this contribution in your paper.

Comment by the author(s): We thank the reviewer for this suggestion. We have mentioned the suggested citation in three different sections of the paper (cited with the number 27 which is Sadik-Zada, E. R., & Gatto, A. (2020). The puzzle of greenhouse gas footprints of oil abundance. Socio-Economic Planning Sciences, 100936). First, we cited it at the end of the literature review in the following paragraph:

“However, a problem arises in natural resource-based countries/regions that are bounded and over-reliant on polluting tradable (e.g., oil and mining) and non-tradable sectors (e.g., construction). Thus, the challenges that these natural-resource regions face are not only related to technological change but to the development of institutional capabilities and policies for the acceleration of structural change towards a low-carbon tertiary sector [26, 27].”

Second, we cited it in the middle of  4.2. “Discussion of results and implications”:

“Overall, the results of a U-shape curve in Greenland are in line with other studies that have analyzed regions with economies that are natural resource-intensive, e.g., [26,27]. The literature on natural resource dependence and the so-called “Dutch disease” stresses how countries…”

And finally, in the concluding section: “The results are in line with other recent quantitative studies on natural resource-dependent economies where their over-reliance on natural resource sectors ultimately have determined the shape of the curve [26, 27].”

Comment by the reviewer:

The authors employ the usual ARDL methodology for their empirical assessment. This is a suitable methodology for the assessment of the EKC-hypothesis. Nevertheless, ARDL does not account for nonlinearities. There are methodologies such as nonlinear ARDL and nonparametric estimators that take account of this confinement of the linear ARDL models. To account for this limitation mention shortly the confinements of the linear and/or parametric estimators. These are especially model misspecification and omitted variable bias.

Response by the author(s):

We very much appreciate the comment. We were not aware of the study from Sadik-Zada, E. R., & Loewenstein, W. (2020). Drivers of CO2-Emissions in Fossil Fuel abundant settings:(Pooled) mean group and nonparametric panel analyses. Energies13(15), 3956. We have now acknowledged it and cited it (with the number 36). The following paragraph was added at the end of subsection 3.3. “Empirical method”:

“The linear specification of the ARDL model assumes the existence of symmetrical effects (i.e., excludes non-linearities) of the explanatory variables on the dependent variable. Some recent panel data studies employing non-parametric estimators, e.g., [36] have found that this assumption (linear specification) could generate issues of model misspecification and omitted variable bias. Yet, assuming normality in the residuals of the above-mentioned functional relationship and aiming to retain a parsimonious estimation we considered the parametric linear estimation approach.”

The author(s) (2021).

Round 2

Reviewer 1 Report

Well done.